# Effects of Rhythm Step Training on Foot and Lower Limb Balance in Children and Adolescents with Flat Feet: A Radiographic Analysis

**DOI:** 10.3390/medicina60091420

**Published:** 2024-08-30

**Authors:** Ji-Myeong Park, Byung-Cho Min, Byeong-Chae Cho, Kyu-Ri Hwang, Myung-Ki Kim, Jeong-Ha Lee, Min-Jun Choi, Hyeon-Hee Kim, Myung-Sung Kang, Kyoung-Bin Min

**Affiliations:** 1Sports Rehabilitation Center, Ain Hospital, 372, Gyeongin-ro, Michuhol-gu, Incheon 22148, Republic of Korea; come90@korea.ac.kr (J.-M.P.); yavanna7@naver.com (B.-C.M.); 2Nowon Samsung Orthopedics, 456 Nohae-ro, Nowon-gu, Seoul 01762, Republic of Korea; bc.jo81@gmail.com (B.-C.C.); hkr8093@naver.com (K.-R.H.); 3Department of Global Sport Studies, Korea University, 2511 Sejong-ro, Jochiwon-eup, Sejong 30019, Republic of Korea; kmk1905@korea.ac.kr; 4Department of Sports and Exercise Medicine, Biomedical Science, Korea University, 2511 Sejong-ro, Jochiwon-eup, Sejong 30019, Republic of Korea; jjungs98@korea.ac.kr (J.-H.L.); qudgns0933@naver.com (M.-J.C.); 5Department of Physical Therapy, Gim Cheon University, 214 Daehak-ro, Gimcheon-si 39528, Gyeongsangbuk-do, Republic of Korea; banchang0@hanmail.net; 6Lotte Healthcare, 300 Olympic-ro, Songpa-gu, Seoul 05551, Republic of Korea; myungsung.kang@lotte.net

**Keywords:** flat feet, rhythm step training, lower limb balance, pediatric rehabilitation, biomechanics

## Abstract

*Background and Objectives:* Owing to the recent reports regarding the efficacy of rhythm step training (RST) in lower limb muscle development and motor skill enhancement, this study aimed to evaluate the effects of RST on foot and lower limb balance in children and adolescents diagnosed with flat feet using radiographic analysis. *Materials and Methods:* A total of 160 children and adolescents diagnosed with flat feet from a hospital in Seoul were randomly assigned to the general flat feet training (GFFT) (*n* = 80) or RST (*n* = 80) group. Patients in both groups exercised for 50 min once a week for 12 weeks. Key variables, such as quadriceps angle (Q-angle), calcaneal pitch angle (CPA), calcaneal–first metatarsal angle (CFMA), and navicular–cuboid overlap ratio (OR) were measured before and after the intervention. *Results:* Significant improvements in Q-angle (*p* < 0.001), CPA (*p* < 0.001), CFMA (*p* < 0.001), and navicular–cuboid OR (*p* < 0.001) were observed in the RST group compared to the GFFT group. RST was found to be more effective in normalizing the biomechanical function of the calcaneus and improving lower limb function. *Conclusions:* RST significantly enhances foot and lower limb balance in children and adolescents with flat feet, suggesting its potential use as an effective intervention for this population. The study did not specifically analyze the effects of various components of rhythm training, such as music, exercise intensity, and frequency, on the outcomes. Further research is needed to determine how each of these elements individually influences the results.

## 1. Introduction

Deformities of the knee joint are common issues among growing children and can lead to severe physical problems if not properly assessed and treated. Typically, normal knee development progresses from a varus to valgus alignment, which stabilizes around the age of 6 years [1,2]. During this developmental process, weakness and imbalance in muscles, tendons, and ligaments, as well as excessive weight, can cause knee instability [3]. Knee instability, particularly when it persists or worsens during growth, can lead to alterations in gait mechanics, such as increased internal rotation of the tibia and valgus alignment of the knee. These changes result in an uneven distribution of weight across the foot, particularly increasing weight-bearing on the medial side. This increased medial pressure can induce pronation of the ankle joint, which in turn exacerbates the development of flat feet by flattening the medial longitudinal arch. As a result, these biomechanical changes can contribute to limitations in physical activity and motor performance in children [4,5]. Moreover, these issues are often difficult to correct naturally during growth, necessitating active intervention to prevent further musculoskeletal complications [6,7].

The foot and ankle are essential for shock absorption, support, and balance during walking [8,9]. Flexible flat feet, a common alignment disorder, disrupts lower limb biomechanics, leading to pain, disability, and potential pronation distortion syndrome [10,11,12,13,14]. Joint flexibility indicates physical stiffness and fatigue, influencing movement and bone growth during development [15,16]. The rising number of children and adolescents seeking treatment for flat feet, from 4030 in 2010 to 13,862 in 2023, highlights ongoing issues with foot and lower limb balance [17].

Previous interventions aimed at correcting lower limb deformities have included specific exercises to induce muscle growth [18]. However, these exercises can sometimes lead to imbalances or new deformities [19,20]. In contrast, jump exercises have shown promise in activating the medial longitudinal arch muscles and improving the flat feet [21]. Incorporating music into step training can enhance physical and cognitive functions, making exercises more engaging and effective [22,23]. Recent studies have highlighted the benefits of rhythm step training (RST) in improving lower limb muscle development and motor skills [24,25]. This study acknowledges that existing research has suggested that Rhythm Step Training (RST) is effective in enhancing physical development and motor skills in children and adolescents. However, there is a significant lack of studies evaluating not only the potential benefits but also the specific impacts of RST on normalizing biomechanical functions. Therefore, this study aims to analyze the effects of RST on muscle development and balance in the feet and lower limbs of children and adolescents with flat feet and to explore the potential for improving lower limb deformities. In particular, this study will focus on how combining Rhythm Step Training with music can reduce the monotony of simple exercise and simultaneously enhance physical functions, investigating whether this approach can yield better results compared to traditional treatment methods.

## 2. Materials and Methods

### 2.1. Participants

This study was approved by the Research Ethics Committee at Gim Cheon University (approval number: GU-202303-HRa-08-P, approval date: 18 July 2023). Participants were recruited at Nowon Samsung Orthopedics in Seoul, Republic of Korea, where a pediatric orthopedic surgeon selected 173 patients diagnosed with flat feet, aged 6 to 13 years, as potential participants. In this study, participants were selected from patients who visited the hospital during the research period. To ensure uniformity in the training program, only one trainer was assigned to guide all participants throughout the study. Of these, 13 individuals were excluded because they or their guardians did not provide written informed consent, resulting in a final sample of 160 participants. None of the final 160 participants met any further exclusion criteria, and all satisfied the study’s inclusion criteria. Participants were randomly allocated to either the general flat feet training (GFFT) (*n* = 80) or RST (*n* = 80) group. We opted for the F test within Analysis of Variance (repeated measures) as our analytical approach. The study sample size was calculated using G*Power (ver. 3.1.9.7, Germany), with a significance level (α) of 0.05, a power (1 − β) of 0.85, and an effect size (*f*) of 0.24. During the study, a dropout rate of approximately 7.5% was observed [(13/173) × 100], resulting in the final sample size of 160 participants. Prior to the experiment, all participants and their guardians were fully informed about the purpose, content, and procedures of the study, and written consent was obtained. The physical characteristics of the study participants are presented in Table 1.

### 2.2. Procedures

The experimental procedures of this study are shown in Figure 1.

### 2.3. Experimental Design and Measurements

This study aimed to compare the effects of GFFT and RST on bilateral foot and lower limb balance. Changes in the quadriceps angle (Q-angle) were assessed using a scanogram, a type of radiographic image, while the degree of flat feet deformity was determined by measuring the calcaneal pitch angle (CPA), calcaneal–first metatarsal angle (CFMA), and navicular-cuboid overlap ratio (OR) through lateral view radiographs of the foot. To ensure accuracy and minimize potential sources of error, all radiographic measurements were consistently performed by a single pediatric orthopedic specialist, thereby maintaining measurement uniformity throughout the study.

### 2.4. Q-Angle Measurement

The Q-angle, or quadriceps angle, is a key indicator of lower limb alignment, particularly in relation to flat feet. An increased Q-angle often results from foot adduction, femoral internal rotation, and tibial internal rotation, which can lead to knee malalignment and contribute to flat feet development. Clinically, Q-angle measurement is crucial for diagnosing and managing flat feet and related conditions, such as knee instability.

In this study, the Q-angle was measured as the angle between the line connecting the patella’s center to the tibial tubercle and the line connecting the patella to the anterior superior iliac spine [26] (Figure 2a). This provides insights into lower limb alignment and its impact on foot and knee biomechanics.

### 2.5. CPA Measurement

The calcaneal pitch angle (CPA) is a clinically significant measure in diagnosing flat feet, as it helps assess the alignment and structure of the foot’s arch. A lower CPA typically indicates a flattened arch, which is characteristic of a flat feet. In this study, the CPA was measured by assessing the angle formed by the lines connecting the most prominent lower part of the calcaneus to the rounded end of the calcaneus and the big toe [27] (Figure 2b).

### 2.6. CFMA Measurement

CFMA measurement is also used for flat feet identification. In this study, the CFMA was measured by assessing the angle at the intersection of the straight line along the first metatarsal and the upper line of the CPA [28] (Figure 2c).

### 2.7. Navicular–Cuboid OR Measurement

The navicular–cuboid OR is important for assessing the alignment and structural relationships between the navicular and cuboid bones, which are critical in maintaining the medial longitudinal arch of the foot. In the context of flat feet, an abnormal OR can indicate a collapse of the arch, leading to biomechanical imbalances in the lower limb. In this study, the navicular–cuboid OR was measured as the ratio of the length of the navicular bone to the length of the overlap between the navicular and cuboid bones [29] (Figure 2d).

### 2.8. GFFT and RST Exercise Programs

Based on the exercise prescription guidelines of the American College of Sports Medicine [30], we conducted GFFT and RST once a week for 50 min over a 12-week period. The exercise program consisted of a 5 min warm-up, a 40 min main exercise, and a 5 min cool-down.

GFFT was developed by building on the research of [31], with specific modifications to better address the needs of children with flat feet. These modifications included incorporating balance exercises that focused on improving proprioception and lower limb alignment, which are crucial for managing flat feet. Additionally, exercises were adjusted to be age-appropriate and to progressively challenge the participants’ stability and coordination.

RST was structured by modifying and refining the program outlined by [24]. These modifications were designed to ensure the exercises were safe and effective for pediatric participants, with an emphasis on rhythm and timing to improve motor coordination and balance. The exercise intensity was gradually increased over the course of the intervention period to ensure the stability of the pediatric participants, reaching a level of “somewhat hard” on the Rating of Perceived Exertion (RPE) scale.

In the RST program, participants were required to perform jump steps in synchronization with a predetermined beat and rhythm, set at 120 beats per minute (BPM). To maintain exercise intensity, participants were instructed to avoid stepping on the lines and to continue jumping without stopping, even if they could not follow the movements accurately. This ensured that the exercise remained challenging and effective, promoting endurance and rhythm consistency. Table 2 presents the details regarding the exercise programs.

### 2.9. Statistical Analysis

Data were analyzed using IBM SPSS Statistics version 22.0 (SPSS Corp., Chicago, CA, USA). Means and standard deviations were calculated for all measurement variables in both groups. A two-way mixed analysis of variance (ANOVA) was performed to analyze the two independent variables, time (pre-intervention and post-intervention) and group (RST and GFFT), during the 12-week training period and to analyze the interaction effect between the two independent variables. Post hoc tests for the interaction effect included paired-sample *t*-tests for pre- and post-differences within groups and independent-sample *t*-tests for differences between groups. The statistical significance level for all analyses was set at *p* < 0.05.

## 3. Results

The pre- and post-intervention changes in the Q-angle, CPA, CFMA, and navicular–cuboid OR for the RST and GFFT groups are presented in Table 3 and Figure 3.

### 3.1. Q-Angle Changes

After the 12-week exercise intervention, the Q-angle showed significant differences in terms of the time of measurement (*p* < 0.001), groups (*p* < 0.001), and the interaction between the time of measurement and groups (*p* < 0.001) in both left and right foot. Post hoc analysis of the interaction effect revealed no significant within-group difference in the GFFT group for the left foot (*p* = 0.634), while a significant difference was observed in the RST group (*p* < 0.001). For the right foot, significant within-group differences were observed in both GFFT (*p* < 0.05) and RST (*p* < 0.001) groups, with a greater Q-angle improvement in the RST group. Furthermore, significant between-group differences were observed for both the left and right foot (*p* < 0.001).

### 3.2. CPA Changes

After the 12-week exercise intervention, the CPA demonstrated significant differences in terms of the time of measurement (*p* < 0.001), groups (*p* < 0.05), and the interaction between the time of measurement and groups (*p* < 0.005) in the left foot. Similarly, the differences were also significant in terms of the time of measurement (*p* < 0.001), groups (*p* < 0.05), and the interaction between the time of measurement and groups (*p* < 0.001) in the right foot. Post-hoc analysis of the interaction effect revealed significant within-group differences in both the GFFT (*p* < 0.005) and RST (*p* < 0.001) groups for the left foot, with greater increases in the RST group. Moreover, for the right foot, significant within-group differences were observed in both the GFFT (*p* < 0.05) and RST (*p* < 0.001) groups, with greater increases in the RST group. Significant between-group differences were also observed for both the left and right foot (*p* < 0.001).

### 3.3. CFMA Changes

After the 12-week exercise intervention, in the left foot, the CFMA showed significant differences in terms of the time of measurement (*p* < 0.05) and the interaction between the time of measurement and groups (*p* < 0.005), but not between groups. Post hoc analysis of the interaction effect revealed no significant within-group difference in the GFFT group (*p* = 0.672). However, a significant within-group difference was observed in the RST group (*p* < 0.001) as well as significant between-group differences (*p* < 0.001). In the right foot, the CFMA showed significant differences in terms of the time of measurement (*p* < 0.05), groups (*p* < 0.001), and the interaction between the time of measurement and groups (*p* < 0.05). Post hoc analysis of the interaction effect revealed that no significant within-group difference in the GFFT group (*p* = 0.291), although a significant within-group difference was observed in the RST group (*p* < 0.001) as well as significant between-group differences (*p* < 0.001).

### 3.4. Navicular–Cuboid OR Changes

After the 12-week exercise intervention, the navicular–cuboid OR demonstrated significant differences in terms of the time of measurement (*p* < 0.001), groups (*p* < 0.05), and the interaction between the time of measurement and groups (*p* < 0.001) in the left foot. Similarly, in the right foot, significant differences were observed in terms of the time of measurement (*p* < 0.005), groups (*p* < 0.001), and the interaction between the time of measurement and groups (*p* < 0.001). Post hoc analysis of the interaction effect revealed significant within-group differences in both the GFFT (*p* < 0.05) and RST (*p* < 0.001) groups for the left and right foot, with greater decreases in the RST group. Significant between-group differences were also observed (*p* < 0.001).

## 4. Discussion

This study aimed to investigate the calcaneal biomechanical function normalization and lower limb functional activity improvement in children and adolescents with flat feet who underwent RST.

### 4.1. Q-Angle Changes

In this study, the Q-angle was measured to identify functional knee issues and observe changes before and after the exercise intervention. The Q-angle is an important indicator for lower limb alignment assessment. It varies with age, gender, and growth status, and the normal values for children and adolescents aged 6 to 13 years range between 10° and 16° [32,33,34,35,36]. Participants in this study had Q-angles outside the normal range of less than 10 degrees and the Q-angle changed as a result of the exercise intervention. The participants in this study had Q-angles outside this normal range, and the exercise intervention triggered Q-angle changes. In the left foot, RST increased the Q-angle from 6.97 ± 1.52° to 8.22 ± 1.39°, whereas GFFT elicited little Q-angle changes, from 6.60 ± 1.53° to 6.53 ± 1.17°. In the right foot, RST increased the Q-angle from 6.81 ± 1.37° to 8.12 ± 1.26°, whereas GFFT decreased the Q-angle from 6.94 ± 1.45° to 6.48 ± 1.45°. This suggests that RST can improve lower limb alignment and reduce the negative effects of flat feet. Previous studies have indicated that the Q-angle can be influenced by muscle imbalances, particularly between the knee extensors and flexors [37], and gluteal muscle strength, which provides stability between the pelvis and femur, thereby controlling lower limb alignment [38]. Additionally, shortened calf muscles or increased stiffness can reduce the Q-angle [39]. Training involving jumping and bounding has been shown to positively affect lower limb muscle structure and joint mechanics [40], and such exercises can strengthen the quadriceps and other supporting muscles, thereby reducing the excessive Q-angle and improving overall knee function [32]. In this study, the greater improvement in the Q-angle observed with RST compared to GFFT may be attributed to the more substantial increases in quadriceps and gluteal muscle strength achieved through the RST training.

### 4.2. CPA Changes

In this study, changes in the CPA, an important indicator for structural foot assessment, were observed before and after the exercise intervention. Generally, the normal CPA range for adults is reportedly between 17° and 32° [41]. However, the anatomical foot structure, which is undergoing development in children and adolescents, can differ from that of adults, based on gender, genetic factors, and physical activity levels. Previous studies have defined flat feet in children and adolescents using a CPA < 20° [42,43] or <23° [44]. In this study, participant CPA values were approximately 19°, indicating the presence of flat feet. Following the 12-week exercise intervention, RST significantly increased the CPA for both feet. For the left foot, RST increased the CPA from 19.03 ± 3.48° to 21.62 ± 2.59°, whereas GFFT showed a smaller CPA increase, from 18.93 ± 1.53° to 19.85 ± 3.31°. For the right foot, RST increased the CPA from 18.55 ± 3.16° to 21.30 ± 3.56°, whereas GFFT decreased the CPA from 19.15 ± 3.45° to 18.54 ± 3.01°. Previous studies have indicated that jumping and plyometric training can maintain the foot arch by strengthening key muscles, such as the tibialis posterior, flexor digitorum longus, flexor hallucis longus, gastrocnemius, and soleus [45]. Neuromuscular control enhancement can also increase foot stability, preventing foot arch collapse [9]. In one study, jump exercises were found to be effective in foot arch improvement by strengthening the support muscles [46]. In this study, RST demonstrated a greater improvement in CPA compared to GFFT. This may be due to the RST regimen incorporating exercises similar to plyometric training, as suggested by previous research [45]. These exercises likely enhanced intrinsic foot muscle strength and neuromuscular control, which are critical for maintaining the foot arch, thereby resulting in a more significant improvement in CPA.

### 4.3. CFMA Changes

In this study, the CFMA was crucial in flat feet deformity evaluation. Research has indicated that abnormal CFMA changes can occur owing to structural foot alterations caused by fractures of the first metatarsal, muscle imbalances, and muscle weakness [47]. Particularly, the weakening of muscles that support the foot arch, such as the tibialis posterior, is relevant [48]. The normal CFMA ranges for men and women are 128.1–136.1° and 129.3–137.4°, respectively [49]. After the 12-week exercise intervention, RST showed significant CFMA improvements for the left foot in terms of interaction effects, although there were no significant between-group differences. Post hoc analysis revealed significant improvements after RST compared with after GFFT. For the right foot, significant changes were observed in all measures. Previous studies have shown that intrinsic foot muscle strengthening exercises can improve the foot arch [45], thereby significantly impacting dynamic posture and balance [50]. Jumping and landing exercises can strengthen intrinsic foot muscles, such as the plantar fascia, abductor hallucis, and flexor hallucis brevis [51]. This study suggests that RST, which includes jumping and landing exercises as indicated in previous research [50], was more effective than GFFT in improving CFMA. This effectiveness is attributed to the strengthening of intrinsic foot muscles and enhancements in dynamic posture facilitated by these exercises

### 4.4. Navicular–Cuboid OR Changes

This study used the navicular–cuboid OR, an important measure for foot structure evaluation which indicates the stability and support of the medial arch of the foot [27]. Navicular–cuboid ORs vary between children, adolescents, and adults. Generally, the normal range for children and adolescents is slightly higher than that for adults owing to ongoing foot development [27,52,53]. The normal navicular to cuboid OR range in children is 47 ± 14% [54]. After the 12-week exercise intervention, RST significantly decreased the ratio for both feet. For the left foot, RST decreased the ratio from 69.48 ± 13.61 to 54.61 ± 13.88, whereas GFFT elicited a smaller ratio decrease, from 67.81 ± 11.70 to 64.17 ± 11.66. For the right foot, RST and GFFT decreased the ratio from 69.02 ± 5.81 to 53.84 ± 12.57 and from 68.00 ± 12.89 to 64.59 ± 10.53 respectively. Previous studies have suggested that an increased navicular–cuboid OR can result from congenital structural abnormalities, foot trauma, weakness of muscles and ligaments around the foot and ankle, or prolonged standing or walking in improper footwear [52,55,56,57,58]. Jump rope exercises performed at a steady rhythm can help strengthen the intrinsic foot muscles that support the arch [46], thereby maintaining proper navicular and cuboid bone alignment [59]. In this study, RST demonstrated greater efficacy in strengthening intrinsic foot muscles compared to GFFT, likely due to the inclusion of exercises performed with a consistent rhythm, such as jump rope, as suggested in previous research [46]. This rhythmic exercise approach may have contributed to better maintenance of proper alignment of the fibular and cuboid bones, thereby leading to improvements in the navicular–cuboid OR.

This study has some limitations. First, the sample size was relatively small, and a long-term effect was not evaluated. Second, it was difficult to maintain the same exercise intensity and frequency for all participants. Future research should gather more data and examine the applicability of RST across different age groups and conditions to confirm its effectiveness in flat feet treatment.

## 5. Conclusions

This study aimed to evaluate the effectiveness of RST in improving lower limb function in children and adolescents with flat feet. After a 12-week intervention period, RST resulted in a significant reduction in the Q-angle in both the left and right feet, a change not observed in the GFFT group. Additionally, both groups showed an increase in CPA, with a more pronounced improvement in the RST group. RST also led to significant improvements in CFMA, and both groups experienced a decrease in navicular–cuboid OR, with a more notable reduction in the RST group. These findings suggest that RST is a promising approach for addressing biomechanical imbalances caused by flat feet.

For future research, it would be beneficial to measure outcomes such as sustained improvements over the long term and the prevention of flat feet recurrence. Additionally, the challenge of maintaining consistent exercise intensity and frequency should be addressed, perhaps by exploring strategies such as personalized training programs or the use of technology to monitor and adjust exercise parameters. Expanding research to explore the applicability of RST across different age groups and other conditions, such as other types of foot deformities, would also provide valuable insights. These recommendations could guide further studies in optimizing and broadening the use of RST as an effective intervention.

## Figures and Tables

**Figure 1 medicina-60-01420-f001:**
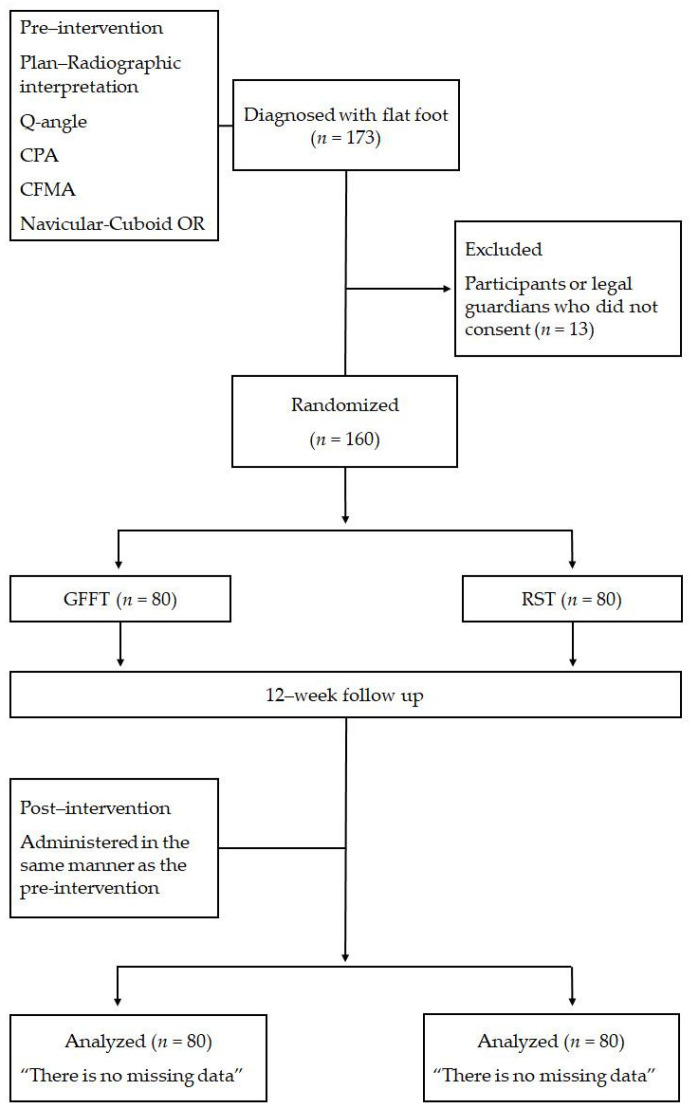
Allocation of participants (flow diagram of the unified criteria for trial reporting). Q-angle: quadriceps angle; CPA: calcaneal pitch angle; CFMA: calcaneal–first metatarsal angle; OR: overlap ratio; GFFT: general flat feet training; RST: rhythm step training.

**Figure 2 medicina-60-01420-f002:**
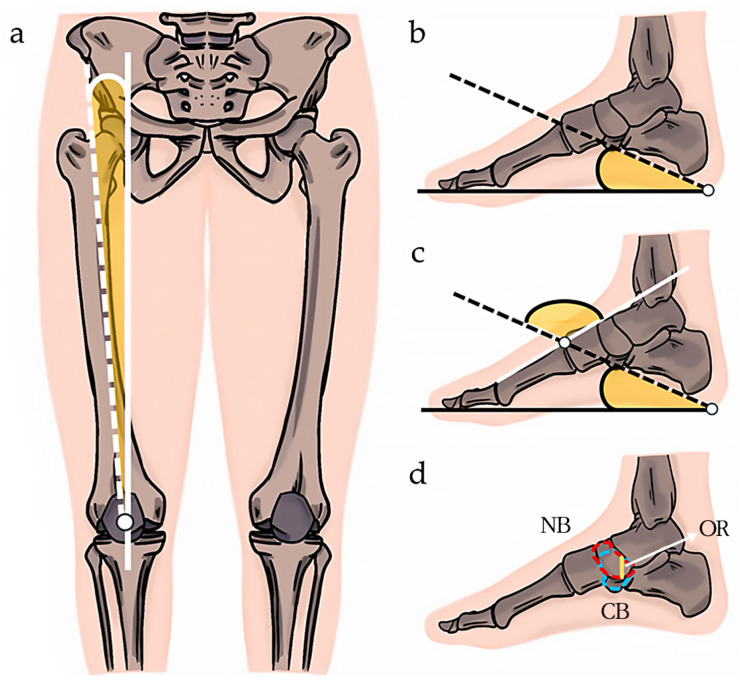
(**a**) Q-angle: the Q-angle is the angle formed by the vertical line drawn upward from the center of the patella and the white dotted line with the anterior superior iliac spine. The white line represents a vertical line upward from the center of the patella, and the white dotted line measures the angle with the anterior superior iliac spine. (**b**) CPA: the black line measures the angle from the rounded end of the calcaneus to the big toe, and the black dotted line measures the angle from the lower protruding part of the calcaneus. (**c**) CFMA: the CFMA is the angle formed at the intersection of the straight line along the first metatarsal and the upper line of the CPA. To measure the CFMA, calculate the angle between the white line, which starts at the first metatarsal and extends toward the tibial direction, and the black dotted line of the CPA, which starts at the calcaneus. (**d**) The navicular–cuboid OR is the ratio of the length of the navicular bone (NB, red) to the length of the overlap between the NB and the cuboid bone (CB, blue). CFMA: calcaneal–first metatarsal angle; CPA: calcaneal pitch angle; OR: overlap ratio.

**Figure 3 medicina-60-01420-f003:**
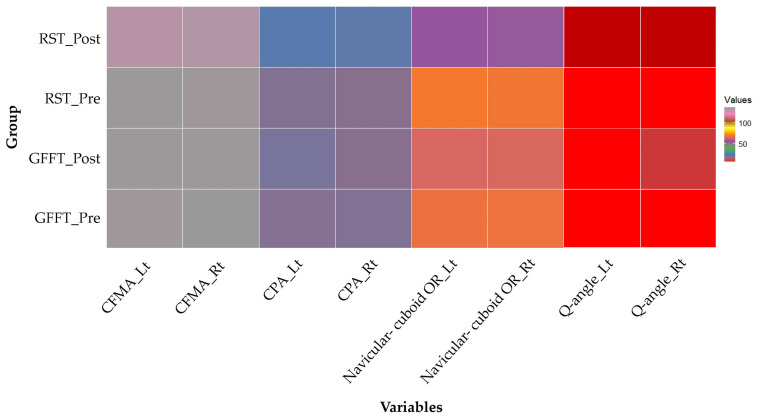
Heatmap visually comparing the values of each variable using colors. Darker hues represent greater values. Q-angle: a significant increase in angle after training is clearly visible in the RST group. CPA: angles increase dramatically following RST, but do not change considerably after GFFT. CFMA: a slight decrease is observed in the RST group, with little changes in the GFFT group. Navicular–cuboid OR: while the ratio declined greatly in the RST group, it also reduced in the GFFT group, albeit less dramatically.

**Table 1 medicina-60-01420-t001:** Participant characteristics.

	Sex	GFFT (*n* = 80)		RST (*n* = 80)	
		M	F	M	F
Variables		40	40	37	43
Age (year)		8.40 ± 1.84	7.88 ± 1.54	8.14 ± 1.95	8.60 ± 2.18
Height (cm)		125.93 ± 10.22	122.18 ± 9.95	126.54 ± 12.90	127.44 ± 14.32
Weight (kg)		30.30 ± 9.87	25.10 ± 6.08	28.86 ± 8.52	28.00 ± 7.00
BMI (kg/m^2^)		15.43 ± 2.51	16.38 ± 2.52	17.84 ± 3.57	17.03 ± 2.67

Data are presented as the mean ± standard deviation. BMI: body mass index; M: male; F: female; GFFT: general flat feet training; RST: rhythm step training.

**Table 2 medicina-60-01420-t002:** GFFT and RST exercise programs.

	Intensity	RPE	Exercise	
			GFFT	RST
Warm-up	1–12 weeks 30 reps × 3 sets	6–9	Foam rolling exercise	
Main Exercise	1–2 weeks (5 s × 10 reps) × 3 sets	10–12	Toe exercise Calf raise Hip cell Hip bridge	Toe exercise Full squat Calf raise Cat and dog core Jump
	3–7 weeks 30 reps × 3 sets		Squat Lunge Balance	Forward and backward Jump step (mat and trek) Side jump step (mat and trek) Cross step (mat and trek) Total jump step (mat and trek)
	8–12 weeks 30 reps × 3 sets		Step box up and down Step box side up and down Step box jumping	One leg forward step Back jump step Turn 180 jump step
Cool-down	1–12 weeks 30 reps × 3 sets	6–9	Achilles tendon stretching and foam rolling exercise	

RPE: rating of perceived exertion; GFFT: general flat feet training; RST: rhythm step training; Reps: repetitions.

**Table 3 medicina-60-01420-t003:** Changes in Q-angle, CPA, CFMA, and navicular–cuboid OR.

	Variables		Group	Pre Intervention	Post Intervention		*F*	*p*
Lower limbs	Q-angle	Lt	GFFT	6.60 ± 1.53	6.53 ± 1.17	G	31.876	0.000 ***
						T	20.973	0.000 ***
			RST	6.97 ± 1.52	8.22 ± 1.39 ^###†††^			
						G × T	26.538	0.000 ***
		Rt	GFFT	6.94 ± 1.45	6.48 ± 1.45 ^†^	G	17.731	0.000 ***
						T	11.534	0.000 ***
			RST	6.81 ± 1.37	8.12 ± 1.26 ^###†††^			
						G × T	49.582	0.000 ***
Arch	CPA	Lt	GFFT	18.93 ± 1.53	19.85 ± 3.31 ^††^	G	4.844	0.029 *
						T	50.327	0.000 ***
			RST	19.03 ± 3.48	21.62 ± 2.59 ^###†††^			
						G × T	11.394	0.001 **
		Rt	GFFT	19.15 ± 3.45	18.54 ± 3.01 ^†^	G	5.642	0.019 *
						T	17.617	0.000 ***
			RST	18.55 ± 3.16	21.30 ± 3.56 ^###†††^			
						G × T	43.623	0.000 ***
	CFMA	Lt	GFFT	135.56 ± 12.20	136.18 ± 5.42	G	3.061	0.082
						T	6.203	0.014 *
			RST	136.51 ± 6.24	131.86 ± 5.90 ^###†††^			
						G × T	10.646	0.001 **
		Rt	GFFT	136.80 ± 6.05	136.19 ± 5.53	G	8.733	0.004 **
						T	15.062	0.000 ***
			RST	135.77 ± 5.81	132.78 ± 4.88 ^###†††^			
						G × T	6.602	0.011 *
	Navicular–cuboid OR	Lt	GFFT	67.81 ± 11.70	64.17 ± 11.66 ^†^	G	5.569	0.020 *
						T	67.296	0.000 ***
			RST	69.48 ± 13.61	54.61 ± 13.88 ^###†††^			
						G × T	24.765	0.000 ***
		Rt	GFFT	68.00 ± 12.89	64.59 ± 10.53 ^†^	G	8.717	0.004 **
						T	84.457	0.000 ***
			RST	69.02 ± 5.81	53.84 ± 12.57 ^###†††^			
						G × T	33.917	0.000 ***

Data are presented as the mean ± standard deviation. Q-angle: quadriceps angle; CPA: calcaneal pitch angle; CFMA: calcaneal–first metatarsal angle; OR: overlap ratio; GFFT: general flat feet training; RST: rhythm step training; G: group; T: time; G × T: group × time; Lt: left foot; Rt: right foot. ^###^ *p* < 0.001: significant difference between two groups. ^†^ *p* < 0.05, ^††^ *p* < 0.005, ^†††^ *p* < 0.001: significant difference between pre- and post-test. * *p* < 0.05, ** *p* < 0.005, *** *p* < 0.001: significant main effect or interaction.

## Data Availability

Study data are included within the article.

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
