# Peer review of "Effects of Rhythm Step Training on Foot and Lower Limb Balance in Children and Adolescents with Flat Feet: A Radiographic Analysis"

_medicina, 2024, doi:10.3390/medicina60091420_

Round 1
Reviewer 1 Report
Comments and Suggestions for Authors
Abstract:
1. Material: State the age range of the participants to provide more context on the study population.
2. Result: Mentioning specific statistical values (e.g., p-values, confidence intervals) could provide more detail and support the significance of the findings. Including a brief statement on the study's limitations or areas for future research could add depth to the conclusions.
3. The abstract does not mention any potential side effects or risks associated with RST, which could be important for understanding the overall safety and feasibility of the intervention.
1. Introduction
1. the section in line 43 to 49 could be enhanced by including more details on the specific mechanisms by which knee instability leads to flat foot.
2. The significance of the study is well-stated, especially regarding the potential benefits of rhythm step training (RST). However, the introduction could better emphasize the gap in the current literature (such as other potential method) that this study aims to address on RST in last paragraph of introduction.
2. Material and methods
2.1 Participants
1. Line 84: “N hospital” kindly confirm the name of hospital.
2. There is some redundancy in mentioning the selection process by pediatricians twice (lines 85-86 and 86-87). This could be consolidated for better readability.
3. Mentioning that participants were selected by pediatricians at a specific hospital provides clarity on the recruitment source, but it would be beneficial to briefly describe any criteria the pediatricians used to select participants diagnosed with flat foot.
4. Refer line 90 -92, it would be beneficial to include a brief explanation of why these specific parameters (α, power, and effect size) were chosen to provide context for the sample size calculation.
5. The exclusion criteria for participants are not explicitly mentioned. Including any specific criteria for exclusion (beyond the lack of consent) would provide a more comprehensive understanding of the participant selection process.
6. Any potential biases introduced by the recruitment process (e.g., selection by pediatricians at a single hospital) should be acknowledged and discussed to provide a balanced view of the study's methodology.
2.2 Procedures
Figure 1 is a flow diagram; however, it does not have the arrow direction of flow. Kindly add the direction of arrow.
2.3 Experimental Design and Measurements
Consistency in terminology is important. Ensure that terms like "radiological images" and "scanogram" are clearly defined and used consistently throughout the section.
2.4 Q–Angle Measurement
1. It would be beneficial to briefly explain the significance of Q-angle in the context of flat foot and lower limb biomechanics. Highlighting the clinical importance of Q-angle measurements in diagnosing and managing flat foot and related conditions would strengthen this section.
2. Describe what is depicted in the figure 2 briefly within the text to provide immediate context.
2.5 CPA Measurement
1. providing some background on the clinical relevance of CPA in flat foot diagnosis would enhance this section. Brief description of the figure 3b within the text would provide better immediate context.
2. It would be beneficial to mention any potential sources of error in CPA measurement and how they are mitigated to ensure accuracy.
2.6 Navicular-Cuboid OR Measurement
1. Brief explanation of why this ratio is important in the context of flat foot and lower limb biomechanics.
2. Discuss any potential sources of error in measuring the navicular-cuboid OR and how these are addressed to ensure accuracy.
2.8 GFFT and RST Exercise Programs
1. Detailed descriptions of these GFFT and SBRT modifications (line 158-159) would help clarify how the program was adapted to suit the study’s needs.
2. 164 and 165 check the spelling.
2.9 Statistical Analysis
1. Justify why this ANOVA method was chosen over others, such as repeated measures ANOVA or MANOVA, given the nature of the study and the variables involved.
3. Result
3.1 Q–Angle Changes:
1. The section clearly distinguishes between within-group and between-group differences. However, it would be beneficial to provide more context or possible reasons for the lack of significant change in the GFFT group for the left foot.
2. Including effect sizes for the differences would help in understanding the practical significance of these findings.
3. The greater improvement in the RST group is noted, but there should be a discussion on why RST might be more effective than GFFT, perhaps referring to underlying mechanisms or theoretical foundations.
3.2. CPA Changes
3. elaborate on the magnitude of CPA changes and their potential impact on clinical outcomes or physical performance.
4. Suggest More detail on the post-hoc analysis results, possibly with graphical representation (e.g., bar charts or line graphs), would enhance clarity.
3.3. CFMA Changes:
1. The reasons for significant changes in the RST group but not in the GFFT group should be hypothesized. Are there specific components of RST that make it more effective?
3.4 Navicular-Cuboid OR Changes:
1. discussing the biomechanical or physiological reasons behind these changes would provide a deeper understanding.
2. The clinical significance of the observed decreases in the navicular-cuboid OR should be discussed. How do these changes translate into improvements in foot function or overall health?
4. Discussion
4.1. Q-Angle Changes
1. However, there is some redundancy in stating that the Q-angle changes as a result of the exercise intervention twice (lines 265–267). Kindly streamline it.
2. The significant decrease in Q-angle with RST is well-highlighted, but the clinical implications of this change could be more thoroughly discussed. How does this improvement translate into functional benefits for the participants?
3. Additionally, discussing any contrasting findings from the literature and offering potential explanations for these discrepancies could provide a more balanced view.
4.3. CFMA Changes
1. The discussion on CFMA changes is comprehensive, but it would benefit from a deeper analysis of the underlying reasons why RST was more effective than GFFT. Are there specific components of RST that are particularly beneficial?
2. How do these changes affect the participants' risk of injury or their overall foot health?
3. Integrating the CFMA changes with other observed outcomes (e.g., Q-angle, CPA) could provide a more holistic view of the benefits of RST. How do these improvements interact to enhance overall lower limb function?
5. Conclusion
1. Specifying what outcomes should be measured in future long-term studies (e.g., sustained improvements, prevention of flat foot recurrence) would provide clearer direction for future research.
2. The challenge of maintaining consistent exercise intensity and frequency is noted. Proposing potential solutions or strategies to address this issue in future studies could be beneficial.
3. Expanding on the applicability of RST across different age groups and conditions is a good suggestion. Providing more detailed recommendations for future research in this area would strengthen the conclusion.

English is good and understandable.
Author Response
We greatly appreciate your time and effort in reviewing this manuscript.
Based on the comments and suggestions received, we have made revisions and incorporated them in the revised manuscript. Additionally, some corrections have been made for typographical and grammatical errors.
Please find attached revised version of the manuscript with changes highlighted in red and point-by-point response to reviewer’s comments as below.
We thank you again for your attention and would like to express our sincerest gratitude to the reviewer’s heartfelt comments.

Reviewer 2 Report
Comments and Suggestions for Authors
Aim of this study was to evaluate the effects of rhythm step training (RST) on foot and lower limb balance in children and adolescents diagnosed with flat foot; authors demonstrated that RST significantly enhances foot and lower limb balance in children and adolescents with flat foot, suggesting its potential use as an effective intervention.
Before publication, authors must revised manuscript:
-Introduction: it would be useful to cut the initial part a little
-Materials and Methods: authors must introduce some sentences on the inclusion and exclusion criteria
-Results and Discussions: good
-References: authors should be inserted a greater number of more recent articles
-Minor revision of English language
Comments on the Quality of English LanguageMinor revision of English language
Author Response
We greatly appreciate your time and effort in reviewing this manuscript.
Based on the comments and suggestions received, we have made revisions and incorporated them in the revised manuscript. Additionally, some corrections have been made for typographical and grammatical errors.
Please find attached revised version of the manuscript with changes highlighted in red and point–by–point response to reviewer’s comments as below.
We thank you again for your attention and would like to express our sincerest gratitude to the reviewer’s heartfelt comments.
Sincerely

Round 2
Reviewer 2 Report
Comments and Suggestions for Authors
Authors improved manuscript according to my review